# A Preliminary Report of Plastic Ingestion by Hawksbill and Green Turtles in the Saudi Arabian Red Sea

**DOI:** 10.3390/ani13020314

**Published:** 2023-01-16

**Authors:** Lyndsey K. Tanabe, Jesse E. M. Cochran, Royale S. Hardenstine, Kirsty Scott, Michael L. Berumen

**Affiliations:** 1Red Sea Research Center, Division of Biological and Environmental Science and Engineering, King Abdullah University of Science and Technology, Thuwal 23955, Saudi Arabia; 2Red Sea Global, Department of Environment and Sustainability, Riyadh 12214, Saudi Arabia

**Keywords:** threat assessment, endangered species, pollution, *Chelonia mydas*, *Eretmochelys imbricata*, marine debris

## Abstract

**Simple Summary:**

Plastic production has increased over the past decades, causing more plastic pollution to enter marine ecosystems. Because plastics persist in the environment for hundreds of years, they have become a threat to many living organisms. Animals may mistake plastic debris for food, which can cause illness or mortality. In this study, we analyzed the gastrointestinal tracts of ten sea turtles to assess the prevalence of plastic ingestion in the Saudi Arabian Red Sea. Plastics > 1 mm were collected and categorized into plastic type and color. This was the first report of plastic ingestion by turtles in the Red Sea. It is important to have baseline data on plastic ingestion because the human population surrounding the Red Sea is expected to increase within the coming decades. With more people residing in coastal areas, there may be an increased amount of plastic debris entering the ocean. This study found that 40% of the turtles had ingested plastics, meaning that plastic is currently a threat to turtles in the region, and conservation efforts should be implemented to remove plastics from the sea and to prevent plastic pollution from entering the environment.

**Abstract:**

(1) Background: Plastic pollution is a major environmental concern confronting marine animals. Sea turtles are considered a bio-indicator of plastic pollution, but there is little information regarding plastic ingestion by turtles in the Red Sea. With large-scale development projects being built along the Saudi Arabian coast, it is important to have a baseline for plastic ingestion before construction is complete. (2) Methods: Ten deceased sea turtles (four hawksbill and six green turtles) were collected along the Saudi Arabian coastline. Necropsies were conducted, and the entire gastrointestinal tracts were extracted and the contents were passed through a 1 mm mesh sieve. (3) Results: We found that 40% of the turtles in this study had ingested plastics. Thread-like plastics were the most common plastic category, and multi-colored was the most prevalent color category. (4) Conclusions: Monitoring of the plastic ingestion by marine megafauna should be conducted as a long-term assessment of the developments’ impacts. Additionally, conservation efforts should be focused on removing plastics (namely ghost nests and fishing lines) from the reefs and reducing the amount of plastic entering the sea.

## 1. Introduction

Globally, most sea turtle species are at risk of extinction [1] due to human impacts, including incidental bycatch, coastal development, direct poaching, boat strikes, and pollution [1,2,3,4]. Ingestion of plastic debris is a potential threat for all sea turtle species [5] at all life stages [6]. An estimated 12 million tons of plastic enters the ocean each year [7], and the production and disposal of plastic used worldwide are projected to increase over the next decades [8].

Plastic pollution in the marine environment is primarily derived from either sea-based sources, such as rubbish dumped from boats, or land-based sources, such as runoff from rivers, wastewater systems, wind-blown litter, and plastic waste left on beaches [9]. Traces of plastic have been found in the entire trophic chain, ranging from zooplankton to cetaceans [10,11], and even in humans [12]. In the marine environment, plastics can absorb pollutants such as heavy metals, thus acting as a vector by facilitating the transport of harmful chemicals to living organisms [13]. However, it is important to note that a single piece of plastic, in general, does not have enough contamination to be responsible for the death of a turtle (e.g., [14]).

Marine plastic pollution also aids invasive species, such as barnacles and mollusks, as a medium for their dispersal [15]. Sea turtles are considered bio-indicators because they are among the first groups of wild animals affected by plastic pollution [16]. Sea turtles are visual feeders with the ability to see color [17], though the role of color in their food preferences is not fully understood [18].

Sea turtle populations in the Red Sea are understudied, as are the threats they face [19]. Increased development along the Saudi Arabian coast [20] could increase anthropogenic contaminants entering into the marine environment, including plastic pollution. Additionally, the human population growth rate around the Red Sea is expected to double in the next 20–30 years [21]. Given the pace and scope of expanded human development in this region, it is important to establish a baseline for plastic ingestion by Red Sea turtles. The aims of this study were to assess the mass of the plastics (>1 mm) ingested by sea turtles and to identify the most common plastic category and color.

## 2. Materials and Methods

Ten deceased green and hawksbill turtles were collected from the Saudi Arabian Red Sea coast (Figure 1). All ten specimens were strandings that were discovered and referred to us by local fishermen, governmental agencies, or giga-project environmental departments. The entire gastrointestinal (GI) tracts of the hawksbill turtles and green turtles were extracted during necropsies. Although no work with live animals was conducted in this study, the research team had approval from King Abdullah University of Science and Technology (KAUST) and from the Institutional Animal Care and Use Committee (IACUC) to work with marine turtles under protocol 19IACUC07. The minimum curved carapace length and width (CCL and CCW, respectively) and plastron length were recorded [22]. The digestive tracts were removed, and metallic clamps were used to contain the ingested materials. The GI tracts were stored at −20 °C until further analysis was possible. For analysis, the GI tracts were thawed out overnight, and the entire contents of each were strained through a 1 mm mesh sieve using fresh water to dilute the GI contents (following [23]). Plastic pieces were retrieved with forceps, cleaned, dried in a Binder incubator for three days at 30 °C, then stored.

For plastic categorization, the INDICIT protocol was followed (https://indicit.cefe.cnrs.fr/indicit-documents/ accessed on 1 October 2022), which includes five categories: (1) sheet-like plastics (e.g., plastic bags), (2) thread-like plastics (e.g., nylon lines or ropes), (3) foams (e.g., polystyrene foam), (4) fragments (e.g., hard plastics), and (5) non-plastic items (e.g., metal or foil). Additionally, the predominant color of the plastic was recorded. Each piece of plastic was photographed and measured (length × width × height) using a Leica IC80 HD stereoscope and weighed using a Mettler Toledo XS205 Dual Range Balance. For plastic fibers that could not be measured individually, the overall mass was recorded. The plastic categories and colors identified were compared by mass for the total amount of plastic retrieved from the turtles as well as compared by mass for each individual turtle. Due to the limited sample size, no statistical tests were computed to compare plastic categories or colors whether spatially or temporally. We used a non-parametric Kendall tau correlation to assess the relationship between the curved carapace lengths of the stranded turtles and the total mass of the plastics ingested using R Studio [24]. This test was used because some of the data did not meet the statistical prerequisites to conduct parametric tests (i.e., our data showed non-normal distribution).

## 3. Results

Ten turtles were necropsied between 2019 and 2022, and of these, four were hawksbill turtles and six were green turtles. Four out of ten turtles (40%) had plastics >1 mm in their digestive tracts, but there was a large range in the number and mass of the plastics in each turtle (Table 1). Of the turtles that ingested plastics, two were green turtles and two were hawksbill turtles. There was no significant correlation between the curved carapace length of the turtles and the mass of the plastics found in their gastrointestinal (GI) tracts (τb = 0.381, *p* = 0.155). The size of plastic items ranged from 0.19 to 351 cm. Of these, 6% were <0.5 cm, 30% were between 0.5 and 2 cm, and 64% were >2 cm.

Turtle G08_2022, a green turtle reported dead in the Al Wajh marina, had more than 10 times as much plastics by mass in its GI tract than any other turtle. Turtle H01_2019 was a hawksbill turtle found in King Abdullah University of Science and Technology (KAUST) that had a single 351 cm piece of nylon fishing line wrapped around its right front flipper, entering through its mouth, passing through its entire GI tract, and ending out of its cloaca. Turtles H05_2021 and G06_2021 both had relatively small amounts of plastic in their digestive tracts (Table 1). Turtle H05_2021 showed signs of bloating syndrome [25] when it was found in Umluj, and it was brought to the Fakieh Aquarium (Jeddah) for rehabilitation. This condition occurs when turtles have gases built up in their GI tracts, increasing buoyancy and preventing sub-surface dives [25]. This turtle excreted a fragment of hard plastic and later died, suggesting that the plastic could have been the cause of the illness.

Out of the five plastic categories, thread-like plastics were the most common. By mass, they comprised 75% of the plastics (Figure 2a). The thread-like category included ropes, nylon fishing lines, small fibers, and other threads. This was followed by sheet-like plastics, which comprised 22% of the plastics by mass. The sheet-like plastic category included plastic bags and other thin flexible plastics. Hard fragments and non-plastic items (e.g., aluminum foil) comprised the remaining 3%. No items in the industrial plastic or foam categories were found, which includes items consisting of Styrofoam.

Next, the proportion of each type of plastic consumed was compared between the four individual turtles that consumed plastic. Thread-like plastics were the dominant type in each turtle (Figure 2c), and turtle H05_2021 was the only turtle with hard fragments in its digestive tract. Turtle G08_2022 had the highest diversity of plastic, with low proportions of fragments and non-plastic items (Figure 2c). As previously mentioned, turtle H01_2019 had a single item in its digestive tract, which was the 351 cm piece of nylon fishing line and which was categorized as “thread-like”.

Fourteen different colors of plastics were identified. Small fibers that were measured together were categorized as “multi-colored”, which was the predominant category, comprising 31% of the colors by mass (Figure 3a). This was followed by white, which was largely composed of the same fiber types that were very common in turtle G08_2022 (Figure 3a). Other commonly found colors were black, blue, and transparent (Figure 3a).

Finally, the proportion of each color of plastic consumed was compared between the four individual turtles that consumed plastic (Figure 3b). Similar trends persisted in this analysis, with turtle G08_2022 having consumed the highest diversity of colors and turtle H01_2019 having consumed the lowest diversity. Turtle H05_2021 had only consumed plastic of two different colors, whereas turtle G06_2021 had consumed plastic of seven distinct colors (Figure 3b).

## 4. Discussion

Previous studies have found that the surface waters of the Saudi Arabian Red Sea have a low concentration of plastic debris compared to all other coastal seas [26]. This lack of surface plastics is hypothesized to be the result of the low contribution of plastic waste into the Red Sea (no river input) and/or fast removal rates of plastic debris from the surface [26]. Given this low level of existing plastic pollution, the relatively high incidence (40%) of plastic ingestion observed in both the hawksbill and green turtles is concerning and warrants further study.

Thread-like plastics, including fibers, ropes, and nylon fishing lines, were the most common type of plastic detected in the stomachs of the affected turtles. Artisanal fishing is common in Saudi Arabian waters, and lost or abandoned fishing gear, such as nets and fishing lines, have been seen on many reefs along the coast (personal observation). Apart from the threat of entanglement, this gear can also break down and be ingested by turtles and other marine fauna. Reduced fishing pressure could help to limit the future input of thread-like plastics into the Red Sea environment, while reef cleanup efforts could remove discarded gear already present. Both methods have the potential to reduce mortality from gear entanglement while also increasing the tourism appeal of local reefs.

There was no clear pattern in the coloring of ingested plastics, though white, multi-colored, and black were most common out of the 14 color categories recorded. We found that the green turtles in our study were ingesting more colors of plastic compared to the hawksbill turtles, but this might be a result of our small sample size. Additional studies are needed to make definite conclusions about the differences between the species regarding plastic ingestion.

A global analysis of plastic ingestion estimated that up to 52% of sea turtles may have ingested plastic debris [27]. This is higher than our findings, where 40% of the turtles included had ingested plastics >1 mm. Furthermore, a study of 464 green turtles necropsied in Texas between 1987 and 2019, 48.7% were found to have ingested plastics [28] (Table 2). The same study also found that plastic ingestion doubled from 1987–1999 (32.5%) to 2019 (65.5%) [28], highlighting the urgency of this issue. Although no work has been published on plastic ingestion by turtles in the Red Sea, there was a study conducted on debris ingestion by green, hawksbill, and olive ridley turtles (*Lepidochelys olivacea*) in the United Arab Emirates [29,30]. Of the stranded turtles included in that study, 85.7% of green turtles and 83.3% of hawksbill turtles contained marine debris, which is a higher rate than we found in the Saudi Arabian Red Sea [29,30] (Table 2).

The small sample size in our study makes interpretation difficult, and there is a need for future studies to expand on this work, especially in the other countries bordering the Red Sea. The current results showed that 50% of the hawksbill turtles (2/4) and 33% of the green turtles (2/6) had ingested anthropogenic marine debris. In the future, it will be useful to have long-term monitoring data on plastic ingestion by turtles in the Red Sea.

## 5. Conclusions

Monitoring of the plastic ingestion by marine megafauna should be conducted as a long-term assessment of the impacts of the upcoming, large-scale developments (locally known as “giga-projects”). Additionally, conservation efforts should be focused on removing plastics (namely ghost nests and fishing lines) from the reefs and reducing the amount of plastic entering the environment. Sea turtles are flagship species that have broad appeal and significance in ecotourism efforts, including in the nascent tourism industry in Saudi Arabia. This highlights the need to conserve the species by reducing the sources of anthropogenic contamination in the environment. This work provides a baseline for plastic ingestion that could be useful for long-term systematic sampling and for assessing whether the problem of plastic ingestion worsens or improves as the giga-projects continue development. These giga-projects have strong environmental standards, so using this work as a baseline would be a good way to assess if they are indeed improving the quality of the marine environment.

## Figures and Tables

**Figure 1 animals-13-00314-f001:**
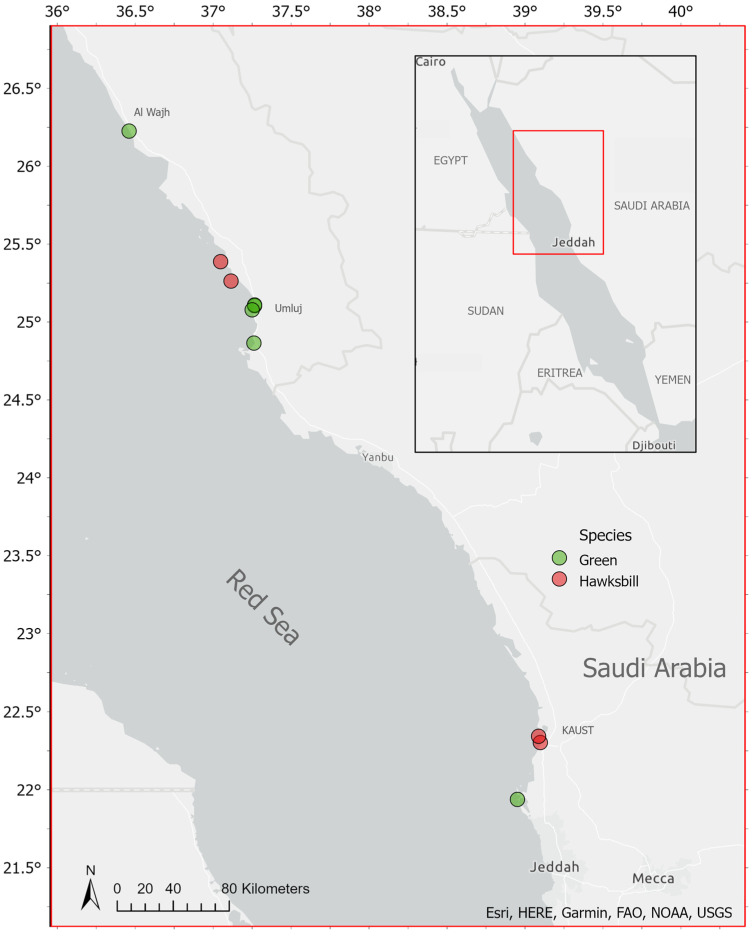
Locations of the stranded turtles collected along the Saudi Arabian Red Sea. Green dots represent the green turtle specimens (*Chelonia mydas*), whereas red dots represent the hawksbill turtle specimens (*Eretmochelys imbricata*).

**Figure 2 animals-13-00314-f002:**
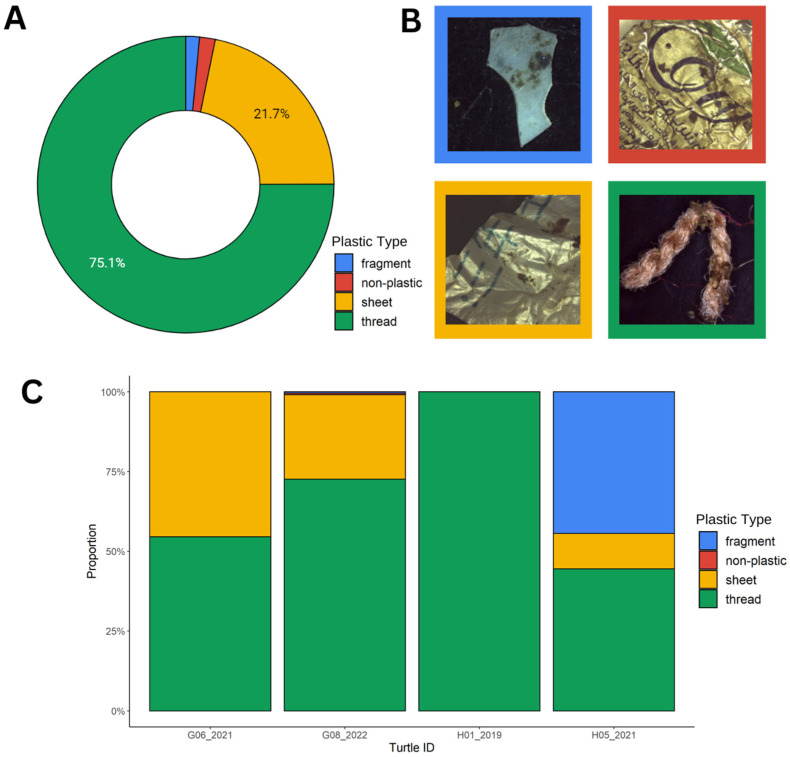
(**A**) The proportion of plastic categories found by mass in the digestive tracts of the turtles in the Saudi Arabian Red Sea. Plastic categories followed the INDICIT protocol, which includes (1) fragments (e.g., hard plastics), (2) non-plastic items (e.g., metal or foil), (3) sheet-like plastics (e.g., plastic bags), (4) thread-like plastics (e.g., nylon lines or ropes), or (5) foams (e.g., polystyrene foam). (**B**) Photographs depicting examples of each type of plastic found in the gastrointestinal contents of the turtles in this study. (**C**) The proportion of each plastic type found in the digestive tracts of the four deceased sea turtles.

**Figure 3 animals-13-00314-f003:**
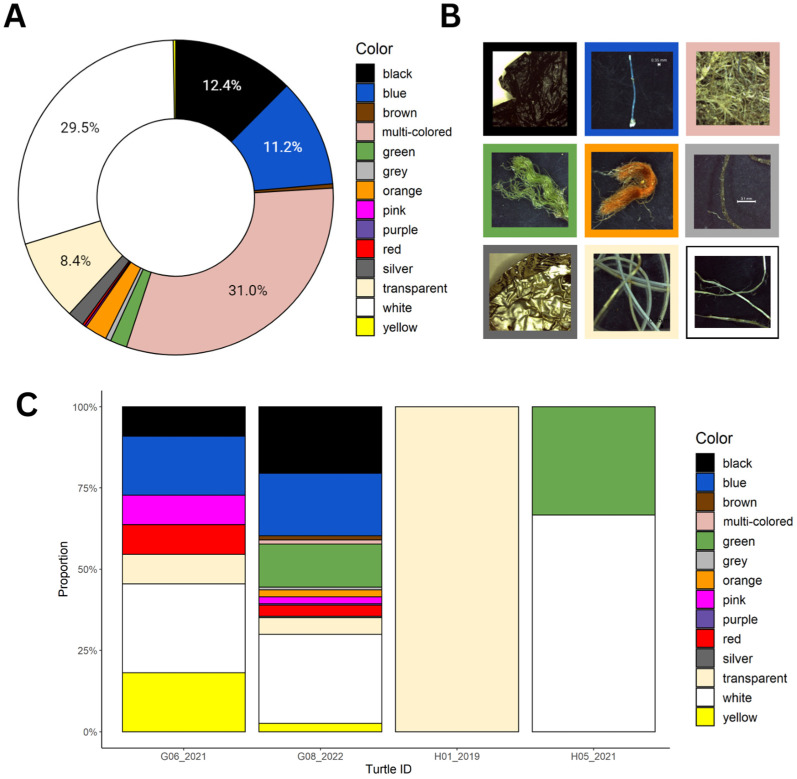
(**A**) The proportion of the colors of plastic found by mass in the digestive tracts of the turtles in the Saudi Arabian Red Sea. (**B**) Photographs displaying examples of the most common colors of plastic found in the gastrointestinal contents of turtles in this study. (**C**) The proportion of each color of plastic found in the digestive tracts of the four deceased sea turtles.

**Table 1 animals-13-00314-t001:** Information on the unique ID, species, location of the deceased turtle, curved carapace length (CCL), and the mass of the plastics (g) found in each turtle’s digestive tract.

Turtle ID	Species	Location	CCL (cm)	Mass of Plastics (g)
H01_2019	Hawksbill	KAUST (Thuwal)	59.6	0.8297
G02_2021	Green	Umluj	74.4	0
G03_2021	Green	Umluj	28.0	0
H05_2021	Hawksbill	Umluj	19.0	0.1747
G06_2021	Green	Jeddah	58.0	0.2194
H07_2021	Hawksbill	Umluj	49.5	0
G08_2022	Green	Al Wajh	78	10.321
G09_2022	Green	Umluj	29.2	0
G10_2022	Green	Umluj	41.3	0
H11_2022	Hawksbill	KAUST (Thuwal)	42.2	0

**Table 2 animals-13-00314-t002:** Comparison of plastic ingestion found in this study with some findings from the literature. All studies used comparable methods (sieving gut contents through a 1 mm mesh sieve) [28,29,30,31,32]. The table includes the location of the study, turtle species, percent (%) of turtles that ingested plastics, and the reference to the study.

Location	Species	% of Turtles that Ingested Plastics	Reference
Saudi Arabian Red Sea	Hawksbill	50% (2/4)	This study
Saudi Arabian Red Sea	Green	33% (2/6)	This study
Texas, USA	Green	48.7% (226/464)	Choi et al., 2021
United Arab Emirates	Green	85.7% (12/14)	Yaghmour et al., 2018
United Arab Emirates	Hawksbill	83.3% (5/6)	Yaghmour et al., 2021
United Arab Emirates	Olive Ridley	28.6% (2/7)	Yaghmour et al., 2021
Azores	Loggerhead	83.3% (20/24)	Pham et al., 2017
Atlantic and Meditteranean	Loggerhead	69.2% (764/1103)	Darmon et al., 2022

## Data Availability

The dataset used and/or analyzed during the current study are available from the corresponding author on request.

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
