# Peer review of "A Preliminary Report of Plastic Ingestion by Hawksbill and Green Turtles in the Saudi Arabian Red Sea"

_animals, 2023, doi:10.3390/ani13020314_

Round 1

Reviewer 1 Report

It is necessary to establish a baseline of plastic ingestion to protect sea turtle. Authors should carefully study the comments and make improvements to the article step by step. After major changes can an article be considered for publication.

1. Please add ethics committee and the approval code to the Materials and Methods section.

2. What are the ways to collect deceased turtles? It is necessary to ensure that this is clear in the Materials and Methods section.

3. Please try to discuss more with literature wherever possible to strengthen your discussion.

4. The time span is quite long, up to four years, so do the different years and locations influence the results? Please discuss it.

5. Table 1 should be three-wire.

6. The major concern is whether such a small sample size is enough to support your conclusion.

Author Response

The reviewers suggestions are below in bolded italics, and my responses are in plain text.

Reviewer 1 comments to authors

It is necessary to establish a baseline of plastic ingestion to protect sea turtle. Authors should carefully study the comments and make improvements to the article step by step. After major changes can an article be considered for publication.

Thank you for your constructive comments, we appreciate your feedback and have incorporated your suggestions into the manuscript.

  1. Please add ethics committee and the approval code to the Materials and Methods section.

Thank you for this comment, this has been added to the methods section, “Although no work with live animals was conducted in this study, the research team had approval from King Abdullah University of Science and Technology (KAUST) Institutional Animal Care and Use Committee (IACUC) to work with marine turtles under protocol 19IACUC07.”

  1. What are the ways to collect deceased turtles? It is necessary to ensure that this is clear in the Materials and Methods section.

We have added a sentence in the Methods section for clarity, “All ten specimens were strandings discovered and referred to us by local fishermen, governmental agencies, or giga-project environmental departments.”

  1. Please try to discuss more with literature wherever possible to strengthen your discussion.

Thank you for this comment, we have added more literature to the Discussion, along with the addition of a table. The text now states, “A global analysis of plastic ingestion estimated that up to 52% of sea turtles may have ingested plastic debris [26]. This is higher than our findings, where 40% of the turtles in-cluded in the study had ingested plastics > 1 mm. Furthermore, of 464 green turtles necropsied in Texas between 1987 and 2019, 48.7% were found to have ingested plas-tics [27]. They also found that plastic ingestion doubled from 1987-1999 (32.5%) to 2019 (65.5%) [27], highlighting the urgency of this issue. Although no work has been published on plastic ingestion by turtles in the Red Sea, there was a study conducted on debris ingestion from green, hawksbill, and olive ridley turtles (Lepidochelys oliva-cea) in the United Arab Emirates [27,28]. Of the stranded turtles included in their study, 85.7% of green turtles and 83.3% of hawksbill turtles contained marine debris, which is a higher rate than found in the Saudi Arabian Red Sea [27,28] (Table 2).”

  1. The time span is quite long, up to four years, so do the different years and locations influence the results? Please discuss it.

We are not comparing plastic ingestion over time because of the small sample size. The goal of our study was not to make any general conclusions but provide the first quantification of plastic ingestion for these species from this region. In this way, we hope that the data helps to provide some form of a baseline, but we are not able to draw any conclusions about temporal or spatial patterns due to the limited sampling size. This has been added to the Methods section, “Due to the limited sample size, no statistical tests were computed to compare plastic categories or colors spatially or temporally. We used a non-parametric Kendal-tau correlation to assess the relationship between the curved carapace length of the stranded turtles and the total mass of plastics ingested on R Studio [22] because some of the data did not meet the statistical prerequisites to conduct parametric tests (i.e., our data showed non-normal distribution).”

  1. Table 1 should be three-wire.

The formatting of this table has been adjusted, thank you for this comment.

  1. The major concern is whether such a small sample size is enough to support your conclusion.

As we have noted in the last paragraph of the Discussion, and as noted above in point #4, we are trying to be transparent and cautious about the limitations of our data due to the small sample size. We have tried to ensure that our conclusions will be understood accordingly. We have also added language to the Methods section to clarify that we are not conducting any statistical analyses (following feedback from Reviewer 2, below). 

Reviewer 2 Report

A preliminary report of plastic ingestion by hawksbill and 2 green turtles in the Saudi Arabian Red Sea by Tanabe et al (2022) is providing plastic ingestion by marine turtles.

Although the sample size is low it is importnat to present such results to show the effects of plasitc pollution to megafauna such as sea turtles.

They followed the protocol of INDICIT which is one of the most important project. I suggest authors to make acomparison of their results which is published recently (Darmon et al 2022, Marine pollution Bulletin), I suggest they also use some information to refer to the methodology by Matiddi et al (2019, JOve).

Although they found no significant relationship with only 10 turtles the test result is not clear. If this is r value or not and P value is >0.05 in any case.

The hawksbill turtles and the green turtles are subject to plastic ingestion in other parts of the world and the comparison of such information would also be important.

Author Response

The reviewers suggestions are below in bolded italics, and my responses are in plain text.

Reviewer 2 comments to authors

Although the sample size is low it is importnat to present such results to show the effects of plasitc pollution to megafauna such as sea turtles.

They followed the protocol of INDICIT which is one of the most important project. I suggest authors to make acomparison of their results which is published recently (Darmon et al 2022, Marine pollution Bulletin), I suggest they also use some information to refer to the methodology by Matiddi et al (2019, JOve).

Thank you for providing feedback on our manuscript. We have added text and a new table to the Discussion section a comparison our findings with other publications including the Darmon et al., 2022 paper as suggested. We have also added the Matiddi et al., 2019 citation in our Methods section.

Although they found no significant relationship with only 10 turtles the test result is not clear. If this is r value or not and P value is >0.05 in any case.

We did not run any statistical analyses due to the limited sample size. We have clarified this in the Methods section, “Due to the limited sample size, no statistical tests were computed to compare plastic categories or colors spatially or temporally. We used a non-parametric Kendal-tau correlation to assess the relationship between the curved carapace length of the stranded turtles and the total mass of plastics ingested on R Studio [22] because some of the data did not meet the statistical prerequisites to conduct parametric tests (i.e., our data showed non-normal distribution).”

The hawksbill turtles and the green turtles are subject to plastic ingestion in other parts of the world and the comparison of such information would also be important.

We agree. As noted above, we have added to the Discussion section additional text to expand on the comparisons with other studies.

Reviewer 3 Report

Plastic pollution is (without any doubt) a matter of concern for all living beings, including marine species and also humans. It is a current issue and, therefore, studies about this subject are important nowadays. Therefore, authors should receive credit for this work. The manuscript is well-organised and I especially like the figures and graphs since they are simple, explanatory and well-presented.

However, I have some concerns regarding this study and, consequently, regarding the manuscript itself. From my perspective, here are the major concerns:

1) Plastic consumption can be a threat for living beings, but sometimes it is presented like a very toxic substance/xenobiotic like heavy metals, pesticides or others. Plastic is an inert material, the consumption of a single piece of plastic just for itself (despite the quantity, shape, additives etc..) is not a case of animal poisoning, from a veterinary perspective. It would be appropriate to clarify, even though most authors and papers on the subject don't clarify this sufficiently. But you can actually do. You have necropsied these animals. What was the cause of death? Was it possible to determine? Was it plastic consumption, which lead to progressive loss of weight? Was it because of the quantity or because of some plastic pieces did some injuries in the GI tract? It is not a problem if it was not the cause determined by the pathologist, but that information is important to be presented. Otherwise, readers from other fields will really not understand this and I think there is general public confusion about this subject. This should be clarified at the introduction of course, but also taken into consideration in all other parts of the article.

2) The number of animals analysed is really small to get proper conclusions regarding the "plastic preferences" of each species, of sea turtles in general or even about the location. I would advise the authors to discuss a little bit more the limitations of their study considering this. Moreover, other aspects of their necropsy reports can also be discussed to complement their ideas.

3) The conclusions can be improved and be more practical and specific. Do you think your study is enough to understand the true impact of plastic pollution in sea turtles' health in this region or implement mitigation strategies? If not, or if would like to deeply study this problem, what would you do differently?

Author Response

The reviewers suggestions are below in bolded italics, and my responses are in plain text.

Reviewer 3 comments to authors

Plastic pollution is (without any doubt) a matter of concern for all living beings, including marine species and also humans. It is a current issue and, therefore, studies about this subject are important nowadays. Therefore, authors should receive credit for this work. The manuscript is well-organised and I especially like the figures and graphs since they are simple, explanatory and well-presented.

However, I have some concerns regarding this study and, consequently, regarding the manuscript itself. From my perspective, here are the major concerns:

1) Plastic consumption can be a threat for living beings, but sometimes it is presented like a very toxic substance/xenobiotic like heavy metals, pesticides or others. Plastic is an inert material, the consumption of a single piece of plastic just for itself (despite the quantity, shape, additives etc..) is not a case of animal poisoning, from a veterinary perspective. It would be appropriate to clarify, even though most authors and papers on the subject don't clarify this sufficiently. But you can actually do. You have necropsied these animals. What was the cause of death? Was it possible to determine? Was it plastic consumption, which lead to progressive loss of weight? Was it because of the quantity or because of some plastic pieces did some injuries in the GI tract? It is not a problem if it was not the cause determined by the pathologist, but that information is important to be presented. Otherwise, readers from other fields will really not understand this and I think there is general public confusion about this subject. This should be clarified at the introduction of course, but also taken into consideration in all other parts of the article.

Thank you for your thoughtful review and questions. First, we agree that clarification about the toxicity of plastics is a useful point to address. We have added mention to the Introduction about the specific threat to turtles that plastic ingestion represents, “In the marine environment, plastics can absorb pollutants such as heavy metals, acting as a vector, and facilitating the transport of harmful chemicals to living organisms [13]. Although it is important to note that in general, a single piece of plastic does not have enough contamination to be responsible for the death of a turtle [e.g., 14].”

In general, our necropsies were unable to determine the direct cause of death for most individuals. We have noted specific cases in which we strongly suspect the plastic was at least a contributing factor to the mortality, but it is difficult to definitively demonstrate this. We hope that the data and samples collected from the necropsies may be useful if specialists are interested to analyze them further. For the purposes of this case study, we are simply limiting our report to the presence and description of the observed plastic ingestion. As noted above in response to Reviewers 1 and 2, and in your own feedback below, we are hesitant to over-analyze our results given the small sample size.

2) The number of animals analysed is really small to get proper conclusions regarding the "plastic preferences" of each species, of sea turtles in general or even about the location. I would advise the authors to discuss a little bit more the limitations of their study considering this. Moreover, other aspects of their necropsy reports can also be discussed to complement their ideas.

As noted above, we have made some changes in the manuscript to be clear about the limitations of the study’s conclusion considering the sample sizes. For our necropsies, we followed a SOP but there were no veterinarians present to conduct a conclusive necropsy analysis, so we do not know the exact cause of death for all the individuals. We have added to the methods that we did not do any statistics on the temporal or spatial patterns of plastic type of color ingested, due to the limited sample size.

3) The conclusions can be improved and be more practical and specific. Do you think your study is enough to understand the true impact of plastic pollution in sea turtles' health in this region or implement mitigation strategies? If not, or if would like to deeply study this problem, what would you do differently?

Thank you for bringing up this concern, we have adjusted the Conclusion section to be more specific, “This work provides a baseline of plastic ingestion that could be useful for the giga-projects to use long-term systematic sampling to assess if the problem of plastic ingestion gets worse or better. These giga-projects have strong environmental standards, so this would be a good way to assess if they are indeed improving the quality of the marine environment.”

Round 2

Reviewer 1 Report

Nevertheless, I still remain suspicious of the results in Abstract " We found that 40% of the turtles in the study had ingested plastics. Thread-like plastics were the most common plastic category and multi-colored was the most prevalent color category. ". I don't think such a small sample size can conclude any reliable and scientific results. Therefore, I challenge the editor to decide wheather that is a serious limitation that is acceptable to the journal standards.

Reviewer 3 Report

Authors followed my recommendations and improved their manuscript significantly. 

The only thing I have noticed to be improved is: please indicate the critical p-value you have considered during the non-parametric Kendal-tau correlation.

I have nothing further to add.